# Inference and control of the nosocomial transmission of methicillin-resistant *Staphylococcus aureus*

Sen Pei[1]*, Flaviano Morone[2], Fredrik Liljeros[3], Hernán Makse[2], Jeffrey L Shaman[1]*

[1]Department of Environmental Health Sciences, Mailman School of Public Health, Columbia University, New York, United States; [2]Levich Institute and Physics Department, City College of New York, New York, United States; [3]Department of Sociology, Stockholm University, Stockholm, Sweden

**Abstract** Methicillin-resistant *Staphylococcus aureus* (MRSA) is a continued threat to human health in both community and healthcare settings. In hospitals, control efforts would benefit from accurate estimation of asymptomatic colonization and infection importation rates from the community. However, developing such estimates remains challenging due to limited observation of colonization and complicated transmission dynamics within hospitals and the community. Here, we develop an inference framework that can estimate these key quantities by combining statistical filtering techniques, an agent-based model, and real-world patient-to-patient contact networks, and use this framework to infer nosocomial transmission and infection importation over an outbreak spanning 6 years in 66 Swedish hospitals. In particular, we identify a small number of patients with disproportionately high risk of colonization. In retrospective control experiments, interventions targeted to these individuals yield a substantial improvement over heuristic strategies informed by number of contacts, length of stay and contact tracing.
DOI: https://doi.org/10.7554/eLife.40977.001

*For correspondence:
sp3449@cumc.columbia.edu (SP);
jls106@cumc.columbia.edu (JLS)

## Introduction

Antimicrobial resistance is a global concern in healthcare systems due to its substantial morbidity and mortality burden and the lack of effective treatment options (*CDC, 2013a*; *Magill et al., 2014*; *WHO, 2018*). Among antibiotic-resistant agents, Methicillin-resistant *Staphylococcus aureus* (MRSA) emerges as one of the most widespread and virulent pathogens (*Grundmann et al., 2006*; *Klevens et al., 2007*; *Klein et al., 2007*; *Jarvis et al., 2012*) and has been highlighted as a leading cause of healthcare-associated infections (HAIs) by the U.S. Centers for Disease Control and Prevention (CDC) (*CDC, 2013b*). Initially confined to healthcare facilities, MRSA has since become increasingly prevalent in the broader population in both the United States and Europe (*Chambers, 2001*; *Naimi et al., 2003*; *Zetola et al., 2005*; *Hetem et al., 2012*; *Kouyos et al., 2013*; *Tosas Auguet et al., 2016*). This entwined transmission among hospitals and the community has obscured understanding of the dynamics and persistence of MRSA. Further, MRSA can colonize patients without symptoms for years, during which it can be transmitted stealthily (*Cooper et al., 2004a*). These epidemiological features have greatly complicated its control and elimination.

The prevalence of MRSA has large variations across different countries. In Europe, a general north-south gradient has been observed, with rare incidence in Scandinavian hospitals and much higher occurrence in Mediterranean hospitals (*Stefani and Varaldo, 2003*; *Tiemersma et al., 2004*; *Johnson, 2011*). In particular, Sweden remains one of the few countries with a low prevalence of MRSA infection (*Stenhem et al., 2006*). A substantial proportion of MRSA cases in Sweden has

**eLife digest** Antibiotic-resistant bacteria like the Methicillin-resistant Staphylococcus aureus (MRSA) can live in people for many years without making them sick. During this time, the bacteria can spread to others who come in contact with the MRSA-infected person. The number of people with stealth MRSA infections living in the community has been increasing. As a result, hospitals may not only be dealing with MRSA infections that originated onsite, but also cases imported from the community. That makes tracking and controlling MRSA infections in hospitals difficult.

Now, Pei et al. show that computer modeling can help identify the role MRSA infections from the community play in hospital outbreaks and test ways to control them. In the experiments, data from an MRSA outbreak that occurred at 66 Swedish hospitals over 6 years were analyzed using statistical methods and computer modeling. This helped to identify patients who were likely colonized with MRSA within the hospital and those who had acquired it in the community. Next, Pei et al. used computer modeling to test what would have happened if these high-risk individuals had received interventions to prevent them from spreading MRSA in the hospital. This showed that targeting individuals at high-risk of a MRSA infection could reduce the spread of MRSA in the hospital.

The computer models developed by Pei et al. may help researchers, clinicians and public health officials working to control the spread of antibiotic resistant bacteria. The model can improve our understanding of how antibiotic resistant bacteria spread in healthcare facilities and may enable the development of more effective strategies to control these pathogens. Infection-control strategies created with this system must first be tested in isolated, real-world settings to verify they work before they can be deployed broadly.

DOI: https://doi.org/10.7554/eLife.40977.002

been imported from abroad due to traveling and healthcare contacts in foreign countries (*Stenhem et al., 2010*; *Larsson et al., 2014*). As a consequence, analyzing MRSA outbreaks in Sweden offers a good opportunity to study the hybrid dynamics of MRSA in hospital settings where both nosocomial transmission and importation occurs.

To facilitate better control of MRSA in hospital settings, several critical questions need to be answered. First, what are the relative roles of nosocomial transmission and infection importation from the community? Public health officials require an accurate assessment of the current force of infection within and into hospitals in order to deploy appropriate containment measures; however, with the increasing prevalence of MRSA in the community, disentangling HAIs from infections imported from the community has become difficult. Second, how many patients are colonized, and who and where are these high-risk individuals? Effective control would benefit from accurate determination of asymptomatic colonization rates in the general population; failure to estimate and account for colonization may result in long-term control issues (*Cooper et al., 2004a*). Although colonized patients can be identified using sequencing methods (*Harris et al., 2010*; *Long et al., 2014*), the expense of these assays limits their application, particularly in underdeveloped countries where MRSA has become endemic. In light of this situation, mathematical modeling offers an alternative approach for locating individuals with a high probability of colonization and guiding the targeted deployment of laboratory testing (*Grundmann and Hellriegel, 2006*; *van Kleef et al., 2013*; *Opatowski et al., 2011*). However, this inference problem is again complicated by the unobserved stealth transmission dynamics that occurs in the highly complex time-varying contact networks of the real world (*Donker et al., 2010*; *Vanhems et al., 2013*; *Jarynowski and Liljeros, 2015*; *Obadia et al., 2015a*; *Obadia et al., 2015b*; *Rocha et al., 2016*; *Nekkab et al., 2017*; *Duval et al., 2018*).

To address these issues, here we develop an agent-based network model-Bayesian inference system for estimating unobserved colonization and importation rates from simple incidence records. We use this system to infer the transmission dynamics of the most commonly diagnosed MRSA strain, UK EMRSA-15 (*Grundmann et al., 2010*; *Das et al., 2013*), from multiple Swedish hospitals (Materials and methods). Key features estimated include the number of infections acquired in hospital and imported from outside, as well as the locations of individuals with a high colonization probability. Such information is crucial for designing cost-effective control measures (*Cooper et al., 2003*;

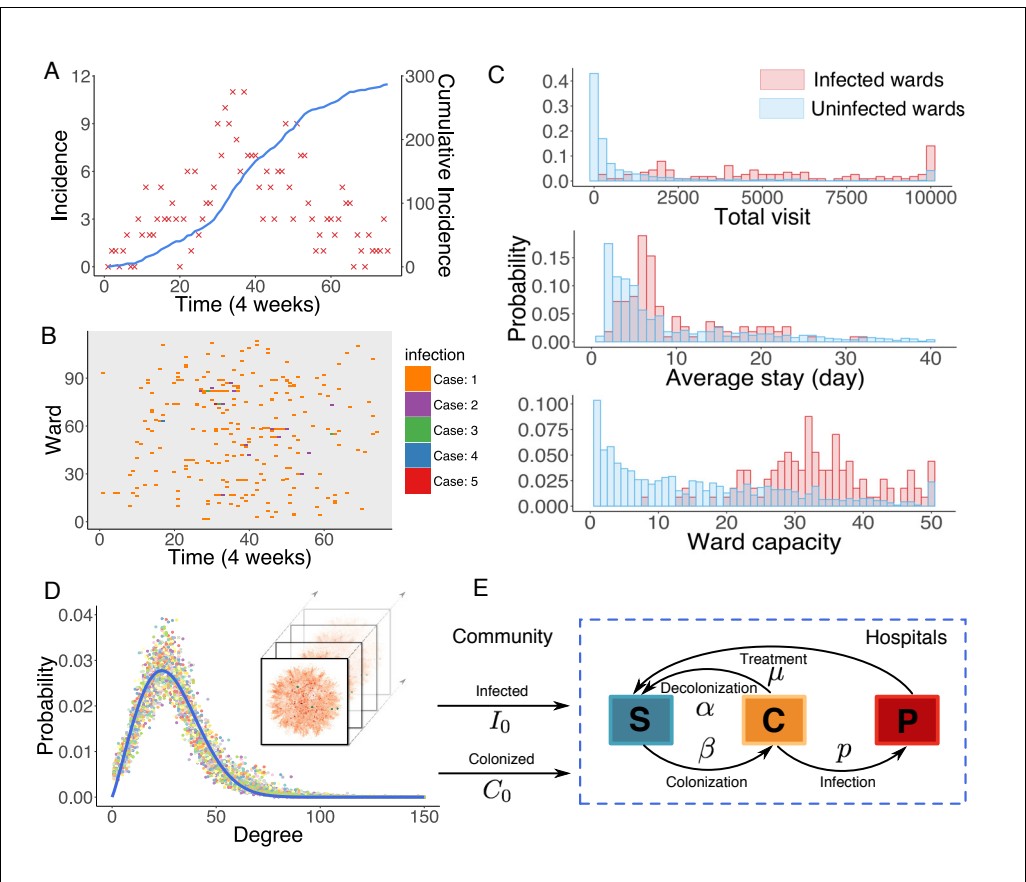

**Figure 1.** Observed incidence of UK EMRSA-15 and the agent-based MRSA transmission model. (**A**) Incidence of UK EMRSA-15 every 4 weeks (red crosses) and cumulative cases (blue curve). (**B**) The raster plot for infections in 114 infected wards. Color indicates number of observed infections during 4-week periods. (**C**) Distributions of total patient numbers per ward (persons, upper panel), patient average length of stay (days, middle panel) and ward capacity (persons, lower panel) for infected and uninfected wards. (**D**) Overlaid degree distributions of 300 weekly aggregated contact networks. The solid blue line is the fitting to a Weibull distribution. Inset shows an illustration of the time-varying contact network. (**E**) A schematic of the model framework. The blue box defines the transmission process within hospitals, and imported infection and colonization from outside the study hospitals are quantified by two parameters $I_0$ and $C_0$.

DOI: https://doi.org/10.7554/eLife.40977.003

The following source data and figure supplements are available for figure 1:

**Source data 1.** Numerical data represented in *Figure 1*.
DOI: https://doi.org/10.7554/eLife.40977.006
**Figure supplement 1.** Association between infection numbers and patient-days per ward.
DOI: https://doi.org/10.7554/eLife.40977.004
**Figure supplement 2.** Patient traffic in the study Swedish hospitals.
DOI: https://doi.org/10.7554/eLife.40977.005

*Cooper et al., 2004b*; *Hubben et al., 2011*; *Worby et al., 2013*). In retrospective control experiments, decolonization of potentially colonized patients outperforms heuristic intervention strategies based on number of contacts, length of stay and contact tracing. These findings indicate that the model-inference system can inform effective, actionable and cost-effective measures for reducing nosocomial transmission.

## Results

### Spatiotemporal features of MRSA infection

Observed incidence over a 6-year outbreak period is reported in *Figure 1A*. The relative date of diagnosis and associated ward of each observed infection were recorded; however, information on type of ward was not provided. Previous studies indicate that heterogeneity in infection risk exists across different types of ward (*Bootsma et al., 2006*). For instance, patients in intensive care units (ICUs) typically suffer a higher risk of infection than those in non-ICU settings due to higher patient-healthcare-worker contact rates, high levels of antibiotic use, and high patient vulnerability to infection. In this study dataset, however, we observed no clear clustering of infections in certain wards. In *Figure 1B*, we use a raster plot to display the distribution of infections in 114 infected wards over time. Infections are distributed without noticeable clustering, presumably due to effective control measures taken in the study hospitals that maintain a low infection rate even in ICU settings (*Tiemersma et al., 2004*).

Distributions of some key statistics of the patient flow in infected wards differ from those in uninfected ones (*Figure 1C*). Infected wards tend to have a higher number of inpatients, a longer average length of stay as well as a larger ward size. Intuitively, the number of MRSA infections in a ward should increase as patient-days within the ward increase. However, the average number of infections is not observed to increase linearly with patient-days, indicating that patient-days per ward alone cannot explain the observed patterns of infection (*Figure 1—figure supplement 1*). While these raw features provide a general understanding of MRSA transmission, they cannot be effectively employed to assess infection or colonization risk in a specific ward due to their largely overlapping distributions, which prevent a clear classification of risk. Instead, a quantitative analysis using mathematical modeling is needed.

### The agent-based model

In hospital settings, MRSA transmission between colonized/infected patients and susceptible individuals is primarily mediated indirectly by healthcare workers (*Lowy, 1998*; *Temime et al., 2009*). As a result, accurate representation of actual contact patterns is crucial for modeling MRSA transmission. Many previous studies have formulated transmission models using ordinary differential equations (ODEs) (*Cooper et al., 2004a*; *Kajita et al., 2007*; *D'Agata et al., 2009*) or stochastic processes (*Forrester et al., 2007*; *Kypraios et al., 2010*). To account for heterogeneity among different settings, several studies have included multiple facilities in a single-model construct, incorporating prior information on facility type in order to characterize and differentiate transmission dynamics (*Bootsma et al., 2006*; *Forrester et al., 2007*). These approaches were then generalized to permit connection among institutions at different scales (hospitals, nursing homes and long-term healthcare facilities, or multiple wards or units within a facility) with time-varying contact patterns (*Thomas et al., 2018*).

In this work, we model nosocomial MRSA transmission using an individual-level agent-based model (*Macal et al., 2014*; *Assab et al., 2017*). One major advantage of using an agent-based approach is that the heterogeneity of contact in different ward types can be accounted for within the model. For instance, even though we were provided no information about ward type, some of the heterogeneity among wards can be represented by the contact pattern specific to each ward, for example a longer length of stay in long-term healthcare units. Other aspects, however, are simplified; for example, due to the observed absence of infection clusters (*Figure 1B*), we assume a uniform transmission rate across different wards within the model. This assumption is ultimately justified by the good agreement between inferred dynamics and observations (as described later).

In the model, transmission occurs on the substrate of a time-varying contact network, which is constructed using the actual hospitalization records from 66 hospitals in Stockholm County, Sweden. In this contact network, nodes represent uniquely labeled patients, connected by undirected links among individuals sharing a ward at a given time. The rationale behind this network construction approach is that, if two patients stay in the same ward simultaneously, the shared healthcare personnel may facilitate transmission between them. The structure of the contact network is relatively stable over time, as indicated by the degree distributions of the weekly aggregated networks (*Figure 1D*). In particular, the degree distributions can be well fitted by a Weibull distribution,

$P(k) = abk^{b-1}e^{-ak^b}$, where $a = 6.15 \times 10^{-4}$ (95% CI: $5.96 \times 10^{-4} - 6.34 \times 10^{-4}$), $b = 2.13$ (95% CI: $2.12 - 2.14$) ($R^2 = 0.95$).

The contact network is time-varying and exhibits high spatiotemporal complexity. The daily in-hospital patient number fluctuates between 4000 and 7000 during the study period (*Figure 1—figure supplement 2A–B*). Patient hospitalization time, readmission time, and patient-to-patient contact time all follow heavy-tailed distributions, spanning several orders of magnitude (*Figure 1—figure supplement 2C*). Moreover, about 100 connected components coexist in the contact network each week (*Figure 1—figure supplement 2D*). Connections between different connected components change over time due to the transfer and readmission of patients. 128,119 patients were transferred from one ward to another during their stay, and another 280,506 patients were readmitted within 1 year of their previous discharge. These patient movements connect healthcare facilities that would otherwise be isolated in the network and facilitate long-range transmission across multiple hospitals. A direct consequence of this patient movement is difficulty tracking the indirect transmission path of MRSA across different hospitals. For instance, a patient located in one hospital can be involved in the transmission occurring in another when he/she moves across multiple facilities. Detailed analyses of the contact network structure and hospitalization traffic can be found in Appendix 1.

Model patients are classified into three categories: susceptible individuals who are free of MRSA ($S$), colonized individuals who carry the bacteria asymptomatically ($C$), and confirmed positive patients ($P$). The model simulates two connected dynamics: nosocomial transmission and importation from the community. Here, the community is broadly defined as all locations outside the study hospitals, and may include households and healthcare facilities not covered in the study. Within hospitals, transitions between states ($S$, $C$, $P$) are governed by parameters that help define either interaction dynamics or the progression of infection. Specifically, a susceptible individual staying in a ward with a colonized person can become MRSA colonized with transmission probability $\beta$ per day. In our model, we assume that patients within a ward have the same rate of contact with each other, presumably mediated by the shared healthcare workers in a ward. The transmission process is density-dependent, as the force of infection in a ward increases with the number of colonized patients within the ward (*Begon et al., 2002*). Upon colonization, asymptomatic persons can return to the susceptible state at a spontaneous decolonization rate $\alpha$, or they can test positive with an infection progression rate $p$. We assume infected patients will receive treatment, no longer spread bacteria, and return to state $S$ with a recovery rate $\mu$. Treatment is assumed to continue until infected patients are clear of MRSA. Given the exponential decay of infection probability, the characteristic treatment period is $1/\mu$ days. Note that colonization only occurs between individuals connected by a link in the contact network, whereas decolonization, infection and recovery progress spontaneously, independent of the contact network. Outside the study hospitals, the transmission process is not explicitly simulated; instead, two additional parameters are introduced to represent transmission intensity. For

**Table 1.** Parameter ranges used in the agent-based transmission model.

| Parameter | Description | Range | Unit |
|---|---|---|---|
| $\alpha$ | Spontaneous decolonization rate | [1/525, 1/175] | per day |
| $p$ | Infection progress rate | [0.1$\alpha$, 0.3$\alpha$] | per day |
| $\mu$ | Recovery rate with treatment | [1/120, 1/20] | per day |
| $\beta$ | Transmission rate in hospitals | [0, 0.01] | per day |
| $I_0$ | Infection importation rate | [0, 0.001] | per admission |
| $C_0$ | Colonization importation rate | [0, 0.1] | per admission |

Sources for parameter ranges – $\alpha$: (*Cooper et al., 2004a*; *Bootsma et al., 2006*; *Eveillard et al., 2006*; *Wang et al., 2013*; *Macal et al., 2014*; *Jarynowski and Liljeros, 2015*); $p$: (*Kajita et al., 2007*; *Jarynowski and Liljeros, 2015*); $\mu$: (*D'Agata et al., 2009*; *Wang et al., 2013*); $\beta$: Prior; $I_0$: Prior; $C_0$: Prior, (*Hidron et al., 2005*; *Eveillard et al., 2006*; *Jarvis et al., 2012*). For each individual, the infection progress rate $p$ is drawn after $\alpha$ is specified.

DOI: https://doi.org/10.7554/eLife.40977.007

patients who appear for the first time in hospital, we assume they belong to states $C$ and $P$ with probability $C_0$ and $I_0$, respectively. As importation rates of colonized and infected patients depend on the time-varying MRSA prevalence outside hospitals, we assume the parameters $C_0$ and $I_0$ are time-dependent. Once patients appear in the contact network, the evolution of their states follows the dynamics as defined above. After discharge, we continue tracking the progression of colonized individuals; however, transmission outside the study hospitals is not represented. The flow of individuals between categories is illustrated schematically in *Figure 1E*.

For a realistic scenario, disease-related model parameters may differ from person to person. To account for this variability during implementation, parameters, for example $\alpha$, $p$ and $\mu$, for each individual are randomly drawn from uniform ranges obtained from prior literature (*Table 1*). The parameter ranges are enlarged slightly to cover the values reported in these works. Our main objective is to infer the three most important parameters governing transmission dynamics: the transmission rate $\beta$, the infection importation rate $I_0$ and the colonization importation rate $C_0$.

## Iterated filtering for agent-based models

To infer epidemiological parameters in an agent-based model, we adapt an iterated filtering (IF) algorithm (*Ionides et al., 2006*; *King et al., 2008*; *Ionides et al., 2011*). IF can be used to infer the maximum likelihood estimates (MLEs) of parameters in epidemic models and has been successfully applied to infectious diseases such as cholera (*King et al., 2008*) and measles (*He et al., 2010*). Initially developed for ODE models, IF has subsequently been generalized for other model forms (e.g. stochastic models) using the plug-and-play approach (*He et al., 2010*). Here, we adapt IF for agent-based models, leveraging an equation-free approach (*Kevrekidis et al., 2003*) that allows for mapping between the system-level observations (e.g. weekly incidence) used for the IF and the individual-level states evolved in the agent-based model (Appendix 1). In applying the IF, we perform multiple iterations using an efficient Bayesian filtering algorithm – the Ensemble Adjustment Kalman Filter (EAKF) (*Anderson, 2001*), which has been widely used in infectious disease forecast and inference (*Shaman and Karspeck, 2012*; *Yang et al., 2015*; *Pei and Shaman, 2017*; *Pei et al., 2018a*; *Kandula et al., 2018*). Details of the IF implementation can be found in Materials and methods.

Before applying the inference system to real-world data, we first need to validate its effectiveness. For the real-world data the inference targets are unobserved, so instead we test the inference system using model-generated synthetic outbreaks for which we know the exact values of the parameter. Although actual MRSA transmission dynamics cannot be fully described by the simplified agent-based model, performing synthetic tests provides validation that the inference system works if the transmission process generally follows the model-specified dynamics.

To generate synthetic outbreak observations, we used the agent-based model to simulate weekly incidence during a one-year period (52 weeks), and then imposed noise to produce the observations used in inference (See details in Appendix 1). We ran 20 iterations of the EAKF within the IF framework. In *Figure 2A*, we display the inference results for the three parameters $\beta$, $I_0$ and $C_0$ at different iterations in one realization of the IF algorithm. The blue horizontal lines mark the target values used to generate the outbreak. The orange boxes show the distribution of posterior parameters (300 ensemble members) after each iteration. The IF algorithm returns the stabilized ensemble mean as the MLEs of parameters. As a result of the stochastic nature of model dynamics and initialization of the inference algorithm, different runs of the IF algorithm usually return slightly different MLEs. To obtain the credible intervals (CIs) for the MLEs, we repeated the inference for 100 times (see Materials and methods). The inferred mean values and 95% CIs for the parameters $\beta$, $I_0$ and $C_0$ are $9.00, [8.07, 9.68] \times 10^{-3}$, $1.91, [1.38, 2.54] \times 10^{-3}$ and $7.18, [5.84, 8.70] \times 10^{-2}$, with the actual values $\beta = 9 \times 10^{-3}$, $I_0 = 2 \times 10^{-3}$ and $C_0 = 7.5 \times 10^{-2}$. The inference system thus accurately estimates $\beta$ and $I_0$ from noisy observations, and slightly underestimates $C_0$. In its implementation, the performance of the inference system depends on the sensitivity of the observations to each parameter. In the agent-based model used here, observed incidence is less sensitive to $C_0$ due to the long period of colonization. As a consequence, estimates of $C_0$ do not always exactly match the actual target and are here biased low. Nevertheless, this slight underestimation does not significantly affect the inferred dynamics. To demonstrate this insensitivity, we ran 1000 simulations using the inferred mean parameters and obtained distributions of weekly incidence from the stochastic agent-based model. The distributions of weekly incidence (blue boxes) are compared with the observed cases

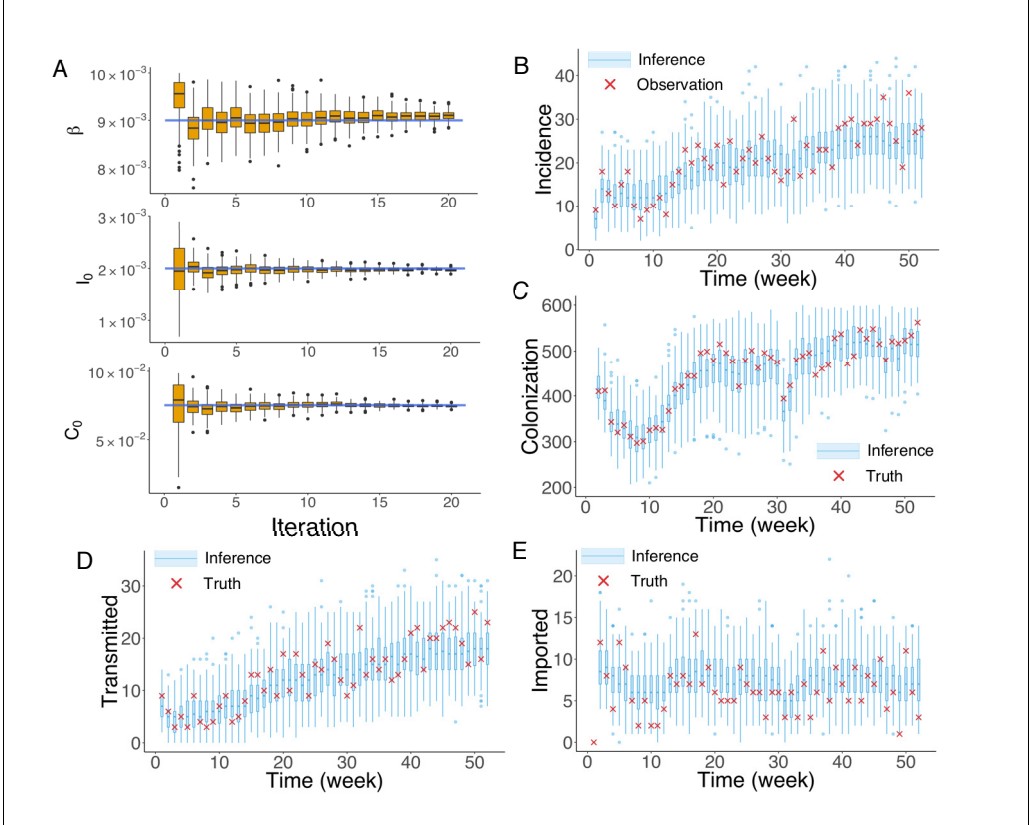

**Figure 2.** Inference of model parameters for a synthetic outbreak. (**A**) Distributions of the posterior parameters $\beta$ (top), $I_0$ (middle) and $C_0$ (bottom) (300 ensemble members) for 20 iterations of inference in one realization of the IF algorithm. Orange Tukey boxes show the median and interquartile (IQR, Q1 to Q3). Whiskers mark the inferred values within the range [Q1-1.5 × IQR, Q3 + 1.5 × IQR]. Dots are outliers. Horizontal blue lines indicate the inference targets used in generating the synthetic outbreak. (**B–C**) Distributions of weekly incidence (**B**) and colonization (**C**) generated from 1000 realizations of simulations using the inferred parameters are shown by the blue boxes. The red crosses represent the synthetic observations used during the inference (**B**) and actual colonization in the outbreak (**C**). (**D–E**) Inference of the transmitted and imported infections. Blue boxes are distributions generated from simulations, and red crosses are the actual values in the synthetic outbreak.

DOI: https://doi.org/10.7554/eLife.40977.008

The following source data and figure supplements are available for figure 2:

**Source data 1.** Numerical data represented in *Figure 2*.
DOI: https://doi.org/10.7554/eLife.40977.014

**Figure supplement 1.** Evaluation of the goodness of fit in *Figure 2B*.
DOI: https://doi.org/10.7554/eLife.40977.009

**Figure supplement 2.** Synthetic test of IF for an outbreak in which the majority of infections are imported.
DOI: https://doi.org/10.7554/eLife.40977.010

**Figure supplement 3.** Evaluation of the goodness of fit in *Figure 2—figure supplement 2B*.
DOI: https://doi.org/10.7554/eLife.40977.011

**Figure supplement 4.** Synthetic test of IF for observations every 4 weeks.
DOI: https://doi.org/10.7554/eLife.40977.012

**Figure supplement 5.** Evaluation of the goodness of fit in *Figure 2—figure supplement 4B*.
DOI: https://doi.org/10.7554/eLife.40977.013

(red crosses) in *Figure 2B*. We also evaluated the agreement between the observed and simulated incidences in *Figure 2B* (*Figure 2—figure supplement 1*; Analysis details are explained in Appendix 1). The inferred dynamics fit the observed incidence well. The Matlab code for synthetic test on an example network is uploaded as an additional file.

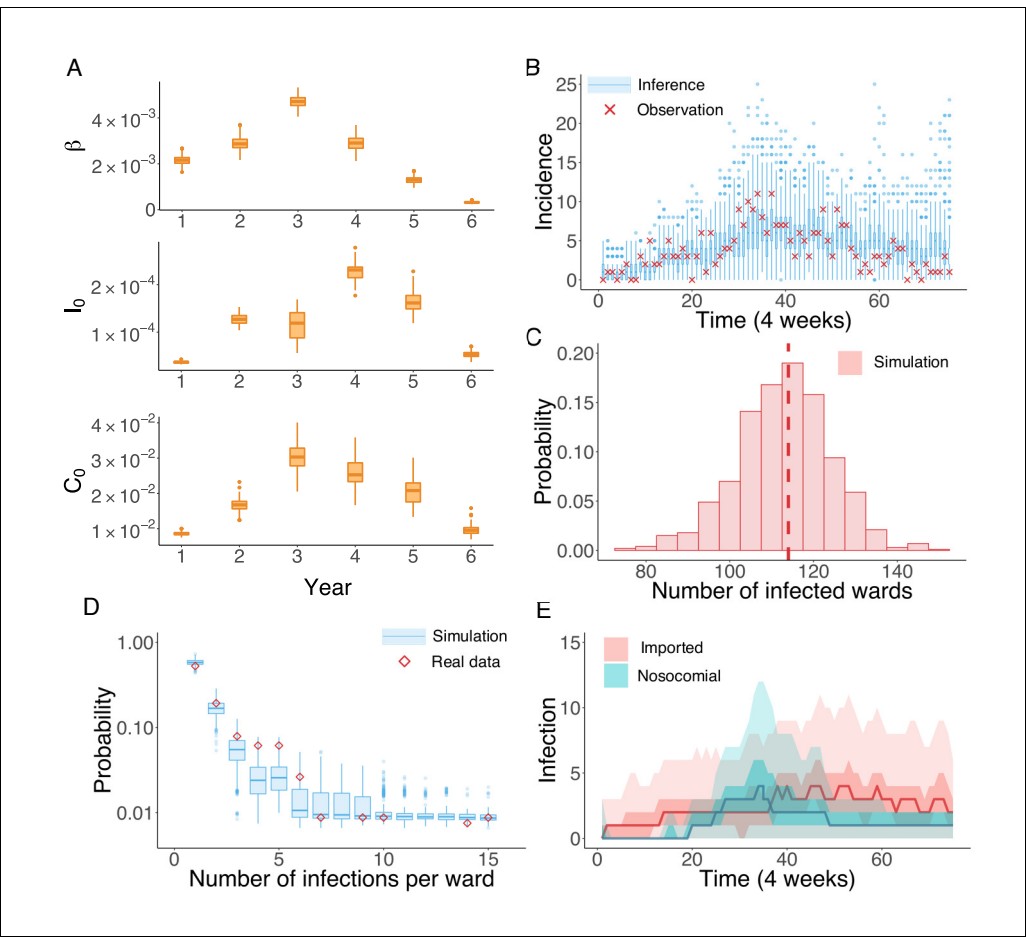

**Figure 3.** Inference of the nosocomial transmission of UK EMRSA-15. (**A**) Inferred distributions of the MLEs for key parameters $\beta$, $I_0$ and $C_0$ over 6 years, obtained from 100 independent realizations of the IF algorithm. (**B**) Observed incidence every 4 weeks (red crosses) and corresponding distributions generated from 1000 simulated outbreaks using the inferred mean parameters (blue boxes and whiskers). (**C**) Distribution of the number of infected wards obtained from 1000 simulations. The vertical red dash line indicates 114, the observed number of infected wards. (**D**) Distributions of the number of infections per ward from 1000 simulations (blue boxes and whiskers). Red diamonds are the observed probabilities. (**E**) Inferred distributions of infections transmitted in hospital (turquoise area) and imported from outside the study hospitals (pink area). The dark areas mark the IQR; light areas show values within the range [Q1-1.5 × IQR, Q3 + 1.5 × IQR].

DOI: https://doi.org/10.7554/eLife.40977.015

The following source data and figure supplements are available for figure 3:

**Source data 1.** Numerical data represented in *Figure 3*.
DOI: https://doi.org/10.7554/eLife.40977.019

**Figure supplement 1.** Distributions of posterior parameters (300 ensemble members) in 20 iterations for different years in one realization of the IF algorithm.
DOI: https://doi.org/10.7554/eLife.40977.016

**Figure supplement 2.** Evaluation of the goodness of fit in *Figure 3B*.
DOI: https://doi.org/10.7554/eLife.40977.017

**Figure supplement 3.** Classification of patients using days from admission to infection.
DOI: https://doi.org/10.7554/eLife.40977.018

We repeated the above analysis for the colonized population (*Figure 2C*) and found that the numbers of unobserved colonized patients can also be well estimated by the inference system. Moreover, the inference system can distinguish the number of infections transmitted in hospital and imported from outside the study hospitals (*Figure 2D–E*). More tests for alternate synthetic

**Table 2.** Inferred parameters and 95% CIs across 6 years using the actual diagnostic data.

| Year | Inferred parameters and 95% CIs | | |
|------|------|------|------|
|  | $\beta$ | $I_0$ | $C_0$ |
| I | $2.16, [1.83, 2.60] \times 10^{-3}$ | $3.67, [3.28, 4.06] \times 10^{-5}$ | $8.61, [7.92, 9.47] \times 10^{-3}$ |
| II | $2.87, [2.48, 3.44] \times 10^{-3}$ | $1.27, [1.13, 1.45] \times 10^{-4}$ | $1.68, [1.40, 1.98] \times 10^{-2}$ |
| III | $4.71, [4.29, 5.13] \times 10^{-3}$ | $6.19, [5.31, 7.48] \times 10^{-5}$ | $3.03, [2.36, 3.62] \times 10^{-2}$ |
| IV | $2.91, [2.47, 3.44] \times 10^{-3}$ | $2.31, [1.93, 2.64] \times 10^{-4}$ | $2.53, [1.85, 3.26] \times 10^{-2}$ |
| V | $3.18, [2.61, 3.79] \times 10^{-4}$ | $1.62, [1.29, 2.04] \times 10^{-4}$ | $2.08, [1.51, 2.63] \times 10^{-2}$ |
| VI | $2.16, [1.83, 2.60] \times 10^{-3}$ | $5.31, [4.27, 6.30] \times 10^{-5}$ | $9.57, [7.72, 12.43] \times 10^{-3}$ |

DOI: https://doi.org/10.7554/eLife.40977.020
The following source data is available for Table 2:
Source data 1. Numerical data represented in *Table 2*.
Results are obtained from 100 independent realizations of the IF algorithm.
DOI: https://doi.org/10.7554/eLife.40977.021

outbreaks and different observation frequencies were performed and are presented in *Figure 2—figure supplements 2–5*.

## Inference of MRSA transmission in swedish hospitals

We next applied the inference system to the UK EMRSA-15 incidence data binned every 4 weeks. Because the UK EMRSA-15 transmission parameters are unlikely to remain constant over the entire 6-year outbreak cycle, we inferred model parameters year by year (52 weeks). Beginning with the first year, we ran the IF inference sequentially through each year. Between 2 consecutive years, the inferred results from the previous year were used to initialize the inference system in the next (see *Figure 3—figure supplement 1*). In *Figure 3A*, we present the distributions of the key parameters $\beta$, $I_0$ and $C_0$ for each year, generated from 100 independent realizations of the IF algorithm. The parameter estimates together with the associated 95% CIs are reported in *Table 2*. All parameter values increased in the first 3 or 4 years, and then gradually decreased thereafter.

The inferred parameters can be plugged back into the model to run simulations and obtain information addressing our questions of interest (see *Video 1* for an example). For instance, we performed 1000 model simulations using the inferred mean parameter values, and generated distributions of incidence from the stochastic agent-based model. These distributions are compared to observations in *Figure 3B*. All observations fall within the whisker range of Tukey boxplots (see more analyses in *Figure 3—figure supplement 2*). To further explore whether some of the key observed statistics can be reproduced using the inferred parameters, we display the distribution of the number of infected wards in *Figure 3C*. The observed number lies at the peak of the simulated distribution (vertical dash line). The spatial distribution of infections among different wards can be characterized by the distribution of wards with a certain number of infections in an outbreak. In *Figure 3D*, we compare this distribution obtained from 1000 simulations with what we observed in the data (red diamonds): the observed distribution agrees well with the

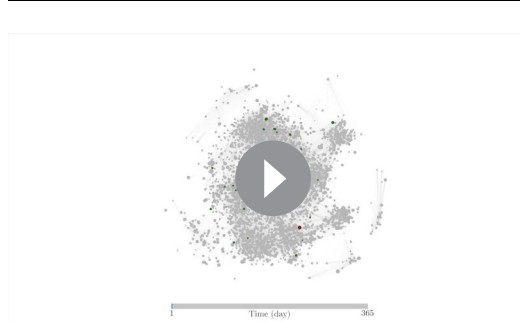

**Video 1.** One realization of the agent-based model simulation. We visualize a single realization of the agent-based model during a one-year period. The grey nodes represent susceptible people, green nodes represent colonized individuals, and red nodes highlight infected patients. The contact network changes from day to day.
DOI: https://doi.org/10.7554/eLife.40977.022

simulated distributions. This close matching indicates that the model structure and inferred parameters can reliably reproduce the observed outbreak pattern in both space and time (see also *Figure 3—figure supplement 2*). In addition to generating a good model fit, the inference system also discriminates the burdens of nosocomial transmission and infection importation. Nosocomial and imported infections are distinguished by the location of MRSA colonization: if patients acquire MRSA in hospital, they are classified as nosocomial transmission cases; otherwise they are imported cases. *Figure 3E* compares the distributions of both types of infections generated from 1000 simulations: a substantial number of infections are inferred as importations. In clinical practice, the number of days between hospital admission and infection is usually used to distinguish hospital-acquired from community-acquired infections, typically with 48 hr used as the threshold. We performed this classification and compared the findings with our inference result. As shown in *Figure 3—figure supplement 3*, the number of imported and nosocomial cases obtained from inference generally matches the classification result using days from admission to infection.

Our findings indicate that, at its onset, during the first year of the outbreak, UK EMRSA-15 gradually invaded the hospital system from the community. Only sporadic nosocomial transmission occurred. With the accumulation of infected and colonized patients in the hospitals, a rise in nosocomial transmission occurred, reflected by an increase of the transmission rate $\beta$ during the third year (*Figure 3A*). Concurrently, both the infection and colonization importation rates, $I_0$ and $C_0$, also experienced growth. This simultaneous rise may have been caused by household transmission initiated by asymptomatically colonized patients discharged from hospitals. After this growth phase,

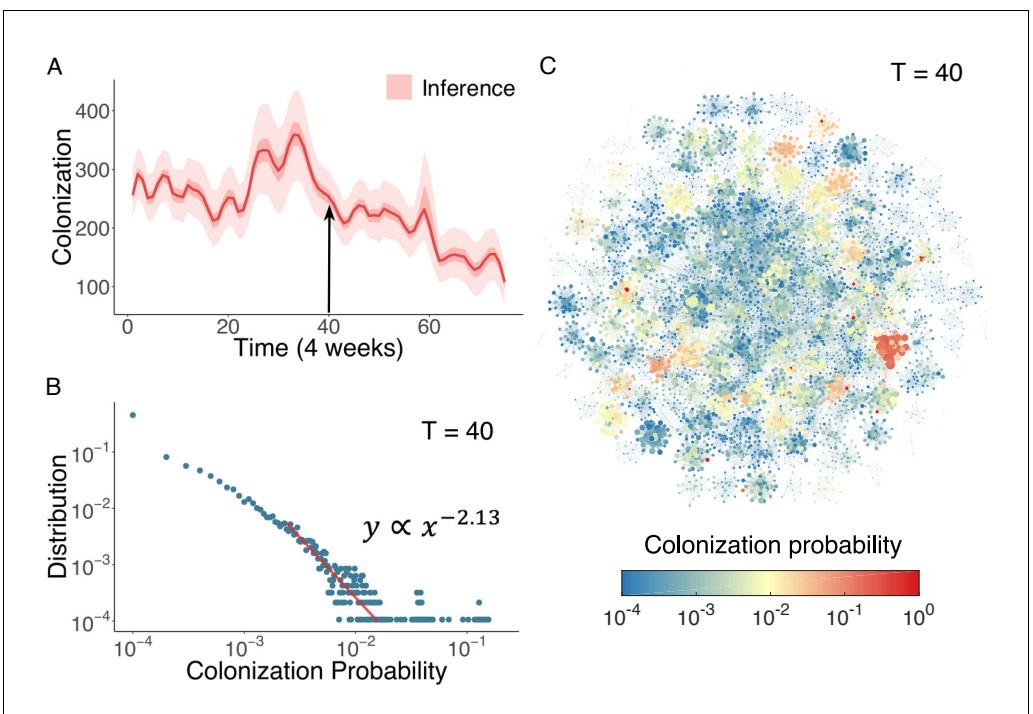

**Figure 4.** Inference of asymptomatic colonization in Swedish hospitals. (**A**) Inferred distributions of colonized patients through time. (**B**) The distribution of colonization probability for each individual in hospital at T = 40 (week 160) calculated from $10^4$ model simulations. The red line is the power-law fitting. (**C**) Visualization of individual-level colonization probability at T = 40. The probability is color-coded in a logarithmic scale. Node size reflects the number of connections.

DOI: https://doi.org/10.7554/eLife.40977.023

The following source data and figure supplement are available for figure 4:

**Source data 1.** Numerical data represented in *Figure 4*.
DOI: https://doi.org/10.7554/eLife.40977.025

**Figure supplement 1.** (A) The KS statistic for different lower bounds of power-law behavior.
DOI: https://doi.org/10.7554/eLife.40977.024

both transmission and importation rates were suppressed. Finally, the UK EMRSA-15 outbreak appears near eliminated in Swedish hospitals, represented by the inferred low values of all parameters. However, if control measures in hospital were to be relaxed, the colonized patients might spark another outbreak due to the lengthy colonization period, which highlights the need for asymptomatic colonization control in order to effect MRSA elimination (*Cooper et al., 2004a*).

## Designing cost-effective interventions

Asymptomatic colonization is a major issue hindering the control and elimination of MRSA in hospitals (*Cooper et al., 2004a*). Screening can identify colonized patients and evaluate the general colonization burden; however, it is an inefficient and costly measure that wastes resources that otherwise could be used to solve more urgent problems. As shown above, given the heterogeneity of contact among patients, levels of exposure to the hazard of colonization differ substantially. As a result, more efficient intervention strategies can be designed that leverage this individual-level heterogeneity.

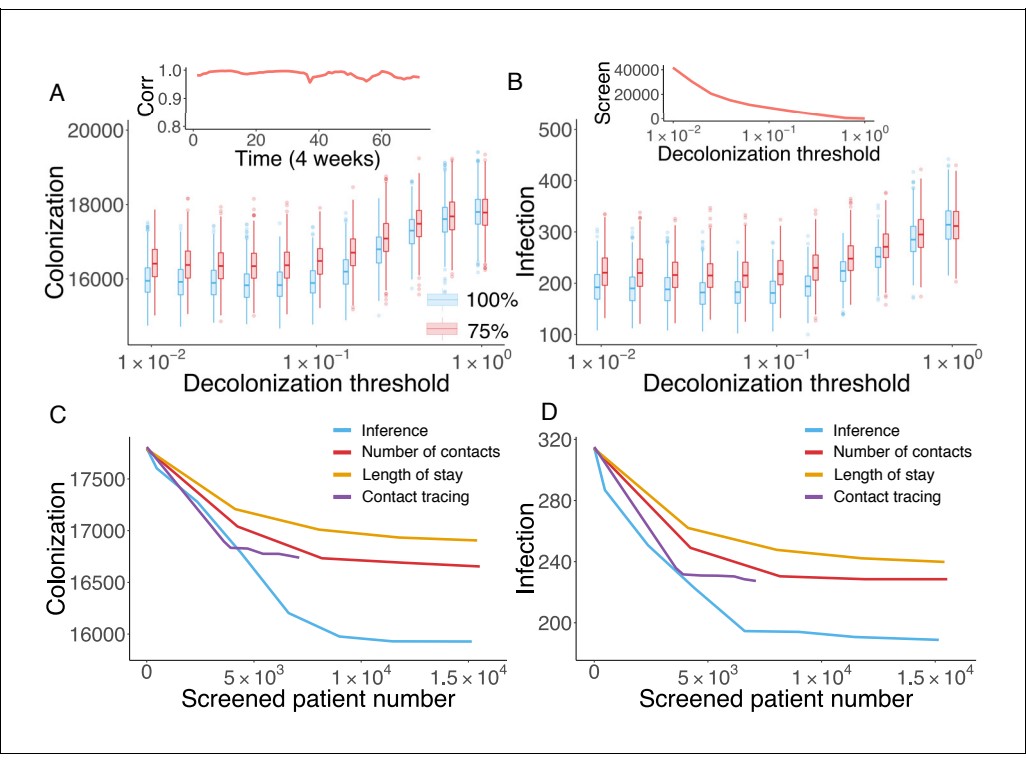

**Figure 5.** Retrospective control experiment in Swedish hospitals. The cumulative cases of colonization (**A**) and infection (**B**) after decolonizing patients with a hazard of colonization higher than a specified decolonization threshold. Simulations were performed with decolonization success rates of 100% (blue boxes) and 75% (red boxes). Distributions were obtained from 1000 realizations of the retrospective control experiment. The inset in (**A**) reports the Pearson correlation coefficient between colonization probability estimated in real time and that obtained using information from the whole course of the epidemic. The inset in (**B**) shows the number of screened patients as a function of the decolonization threshold. (**C–D**) Comparison of the inference-based intervention with heuristic control measures informed by number of contacts, length of stay and contact tracing. Curves are average cumulative cases obtained from 1000 experiments with a 100% decolonization success rate.

DOI: https://doi.org/10.7554/eLife.40977.026

The following source data is available for figure 5:

**Source data 1.** Numerical data represented in *Figure 5*.
DOI: https://doi.org/10.7554/eLife.40977.027

In *Figure 4A*, we display the inferred distribution of colonized patients in the Swedish hospitals over time. Colonized patient numbers peak in the middle of the record and decline thereafter. To determine who and where these high-risk individuals reside within the network, we can use the agent-based model to quantify colonization risk at the individual level. The distribution of individual colonization probability at T = 40 (week 160), generated from $10^4$ simulations using inferred parameters, is displayed in *Figure 4B*. A clear heavy-tailed power-law distribution $y \propto x^{-2.13}$ emerges in which the colonization probability spans several orders of magnitude (see *Figure 4—figure supplement 1* and Appendix 1 for a rigorous statistical analysis of this distribution) (*Clauset et al., 2009*; *Muchnik et al., 2013*). The complex spatiotemporal interaction patterns within the network give rise to a small number of patients with a disproportionately high risk of colonization. To examine how these individuals distribute among hospitals, we visualize the colonization probability in *Figure 4C*. High-risk patients tend to appear in densely connected clusters.

Cost-effective interventions can be practiced by the targeted screen and decolonization of identified high-risk patients. In order to evaluate the effectiveness of such interventions, we performed a retrospective control experiment. Specifically, we used the inferred parameters in *Figure 3A* to run the model for 6 years to reproduce the outbreak. Every 4 weeks, we used currently available information (as would be available in real time) to estimate patient colonization probabilities (see details in Materials and methods). The colonization probabilities estimated in real time are highly correlated with the results obtained using information from the whole course of the epidemic, shown in *Figure 4C*. During the model integration, every 4 weeks, we selected patients with an estimated colonization probability higher than a certain threshold for screening. If positive, these inpatients were decolonized. To assess the impact of decolonization success rate on intervention impact, two efficiencies, 100% and 75%, were tested, and we repeated the experiment 1000 times. The findings show that the proposed intervention strategy can avert considerable numbers of colonization and infection (*Figure 5A–B*). Decreasing the decolonization threshold leads to a larger screened population (as shown in the inset of *Figure 5B*), and thus reduces colonization and infection further. However, the marginal benefit becomes negligible below a certain threshold value, as the remaining colonized and infected patients are possibly caused by importation, which cannot be directly controlled by inpatient intervention. The decolonization success rate also plays an important role, as indicated by the increased colonization and infection for the lower success rate.

The advantage of the proposed inference-based intervention can be better appreciated by examining its additional benefit over other heuristic control measures. Here, we compare the performance of the inference-based intervention with three alternative screening strategies informed by patient number of contacts, length of stay and contact tracing. For the former two, at each month, we ranked patients by their current total number of contacts (i.e. cumulative number of connections in the time-varying network since admission) or length of stay in a descending order, and created the screening list using the top-ranked patients. By varying the fraction of patients selected from the ranking (from 0% to 5%), we can inspect the control results for different numbers of screened patients. For contact tracing, upon each observation of infection, we tracked patients who stayed in the same ward with an infected individual within a certain time window prior to the infection, and screened those possibly colonized patients in hospitals. Tracing time windows ranging from 1 day to 14 days were tested. The number of screened patients does not increase significantly with tracing times longer than 14 days. Note that, screening and decolonization are performed only within hospitals. If patients listed for screening have already discharged before the diagnosis of infection, they are screened upon their next re-admission.

In *Figure 5C–D*, the average numbers of colonized and infected patients are compared based on the number of screened patients. Heuristic control measures relying on the number of contacts, length of stay and contact tracing all limit MRSA transmission; however, a substantial additional reduction in both colonization and infection can be achieved through inference-based intervention. On average, inference-based screening of approximately 0.89% (6,617/743,599) of all patients can avert up to 38% (121/315) of infections and 9% (1,610/17,810) of colonizations. In comparison, the other three methods given similar numbers of screened patients only reduced infections and colonizations by 21% and 4% (number of contacts), 27% and 6% (length of stay), and 28% and 6% (contact tracing), respectively.

The colonization probability obtained from inference quantifies individual systemic risk given the general situation of transmission, regardless of the specific location of undetected colonization. In

contrast, screening based on contact tracing identifies colonized individuals related to observed infections; however, with an unknown amount of imported colonization, this approach may overlook a considerable number of colonized patients, who can sustain subsequent transmission. As a result, the inference-based intervention can identify and treat the pivotal individuals, or superspreaders (*Pei and Makse, 2013*; *Pei et al., 2014*; *Pei et al., 2017*; *Pei et al., 2018b*; *Teng et al., 2016*), who may otherwise transmit MRSA asymptomatically in the first place. This preventive approach is more effective than contact tracing in the presence of frequent importation, as it disrupts probable transmission pathways. In real-world hospital settings, the proposed inference-based intervention could be implemented and evaluated in real time: it only requires hospitalization records and ward information.

## Discussion

In this work, we have developed an agent-based model-inference framework that can estimate nosocomial MRSA transmission dynamics in the presence of importation. Further, we have shown that these inferred dynamics can be used to quantify patient colonization risk and guide more effective interventions.

The transmission dynamics generated using the agent-based model are intrinsically stochastic, that is, the observed record of UK EMRSA-15 infections is just one realization among an ensemble of all possible outcomes of an underlying highly stochastic process. In order to evaluate the general risk of MRSA transmission, key epidemiological parameters were inferred from the single observed realization. Previous studies have developed methods to infer transmission risk factors and reconstruct transmission paths using individual-level infection data for diseases such as H1N1 and MERS-CoV (*Cauchemez et al., 2011*; *Cauchemez and Ferguson, 2012*; *Cauchemez et al., 2016*). In particular, Bayesian data augmentation approaches have been applied to MRSA models (*Forrester et al., 2007*; *Kypraios et al., 2010*); however, these approaches are not readily applicable to our dataset. The data assimilation scheme we developed here enables estimation of epidemiological parameters and key transmission information using aggregated incidence data. As demonstrated in the retrospective control experiment, assessment of individual colonization risk using aggregated data can be quite useful for preventing future MRSA transmission, especially when stealth importations are frequent.

In this study, we omitted representation of heterogeneity across different wards. This simplification is valid for the study Swedish hospitals, as we observed no infection clusters and the model reproduced key statistics of observations well. However, in other settings, clustering analysis and ward information may be necessary before the application of the inference system. Should certain wards suffer a much higher rate of infection, a separate suite of parameters can be defined and inferred for these wards, using priors that better represent this more intense transmission. We also only considered transmission among patients staying in the same ward. In the future, more contact information such as healthcare workers shared by a group of patients could be incorporated into the contact network. In addition, as the community defined in the model may include non-sampled hospitals, inferred community risk may have been overestimated as it also included contributions from those healthcare facilities outside the network.

We note that the reported inference results are obtained using only hospitalization records and UK EMRSA-15 case numbers. Should more data (e.g. surgery or treatment records) become available, this additional information could be incorporated into the agent-based model and used to refine the present results. Our model-inference framework provides a foundational platform for flexible simulation and inference of antibiotic resistant pathogens. In this study, we applied this system to Swedish hospitals with low MRSA prevalence. However, in the future, it could be used to provide actionable information for disease control in less developed settings where MRSA is endemic. In a highly interconnected area, transmission of antibiotic resistant pathogens from endemic regions to epidemic-free hospitals is more likely. This risk calls for containment measures in the general population and collaborative control efforts among multiple healthcare facilities (*Smith et al., 2005*; *D'Agata et al., 2009*; *Ciccolini et al., 2014*; *Slayton et al., 2015*).

**Table 3.** Inferred parameters and 95% CIs for three synthetic tests.

| | $\beta$ | $I_0$ | $C_0$ |
|---|---|---|---|
| **Actual** | $9 \times 10^{-3}$ | $2 \times 10^{-3}$ | $7.5 \times 10^{-2}$ |
| **Inference (weekly)** | $9.00, [8.07, 9.68] \times 10^{-3}$ | $1.91, [1.38, 2.54] \times 10^{-3}$ | $7.18, [5.84, 8.70] \times 10^{-2}$ |
| **Actual** | $6 \times 10^{-3}$ | $2 \times 10^{-3}$ | $7.5 \times 10^{-2}$ |
| **Inference (weekly)** | $5.54, [4.17, 5.80] \times 10^{-3}$ | $2.11, [1.52, 2.55] \times 10^{-3}$ | $7.05, [5.79, 8.11] \times 10^{-2}$ |
| **Actual** | $9 \times 10^{-3}$ | $2 \times 10^{-3}$ | $7.5 \times 10^{-2}$ |
| **Inference (monthly)** | $9.00, [8.17, 9.66] \times 10^{-3}$ | $1.99, [1.21, 2.64] \times 10^{-3}$ | $7.14, [5.99, 9.04] \times 10^{-2}$ |

DOI: https://doi.org/10.7554/eLife.40977.029

# Materials and methods

## Data description

The dataset contains admission and discharge records of 743,599 distinct patients from 66 hospitals (271 clinics, 1041 wards) in Stockholm County, Sweden (*Jarynowski and Liljeros, 2015*; *Rocha et al., 2016*), spanning over 3500 continuous days during the 2000s. The exact dates and ward types are confidential for the protection of patient privacy. In total, 2,041,531 admission records were collected. The hospitalization dataset is quite comprehensive as the patients constitute over one third of the total 2.2 million population of Stockholm County. In addition, the dataset also contains individual diagnostic records of MRSA, which provide relative date of diagnosis and the strain of MRSA. Diagnosis was performed on patients with symptomatic infections as well as asymptomatic patients in contact with positive cases. A total of 991 positive cases from 172 different strains were confirmed, and the most prevalent strain was UK EMRSA-15 (289 cases). UK EMRSA-15 is present in 16 countries worldwide (*Grundmann et al., 2010*; *Das et al., 2013*). Here, we focus on this specific strain. Although the dataset spans over 3500 days (nearly 10 years), we limit our study to a 6-year (300-week) period with reported UK EMRSA-15 incidence. We display time series of 4-week incidence and cumulative incidence for UK EMRSA-15 in *Figure 1A*.

## Iterated filtering for agent-based models

We infer system epidemiological parameters using an iterated filtering (IF) algorithm (*Ionides et al., 2006*; *King et al., 2008*; *Ionides et al., 2011*). This algorithm has been coupled with ODE models and used to infer latent variables associated with the transmission of cholera (*King et al., 2008*) and measles (*He et al., 2010*). The IF framework is designed as follows: an ensemble of system states, which represent the distribution of parameters, are repeatedly adjusted using filtering techniques in a series of iterations, during which the variance of the parameters is gradually tuned down. In the process, the distribution of parameters is iteratively optimized per observations and narrowed down to values that achieve maximum likelihood. This approach is based on an analytical proof that guarantees its convergence under mild assumptions (*Ionides et al., 2006*).

In its original implementation, the data assimilation method used in IF is sequential Monte Carlo, or particle filtering (*Arulampalam et al., 2002*). Here, due to the high computational cost of the agent-based model, we use a different efficient data assimilation algorithm - the Ensemble Adjustment Kalman Filter (EAKF) (*Anderson, 2001*). Unlike particle filtering, which requires a large ensemble size (usually of the order $O(10^4)$ or higher) (*Snyder et al., 2008*), the EAKF can generate results similar in performance using only hundreds of ensemble members (*Shaman and Karspeck, 2012*). Originally developed for use in weather prediction, the EAKF assumes a Gaussian distribution of both the prior and likelihood, and adjusts the prior distribution to a posterior using Bayes rule in a deterministic way such that the first two moments (mean and variance) of an observed variable are adjusted while higher moments remain unchanged during the update (*Anderson, 2001*). In epidemiological studies, the EAKF has been widely used for parameter inference and forecast of infectious diseases (*Shaman and Karspeck, 2012*; *Yang et al., 2015*; *Pei and Shaman, 2017*; *Pei et al.,*

2018a; Kandula et al., 2018). The implementation details of the EAKF are introduced in Appendix 1.

In this study, we focus on the inference of three transmission-related parameters: the nosocomial transmission rate $\beta$, the infection importation rate $I_0$ and the colonization importation rate $C_0$. The initial prior ranges for these parameters are reported in *Table 1*. Other disease-related parameters, for example the spontaneous decolonization rate $\alpha$, the infection progress rate $p$, and the recovery rate $\mu$, are drawn uniformly from ranges obtained from previous studies for each individual in the agent-based model (see *Table 1*). Should more specific information about these parameters become available, it may be possible in the future to better constrain the model with their incorporation into the system. In synthetic testing of the IF-EAKF algorithm, we use weekly incidence as observations. Given the parameter vector, $\mathbf{z} = (\beta, I_0, C_0)^T$, the IF-EAKF algorithm proceeds per the pseudo-code in Algorithm 1. During the EAKF update, only the parameters $\beta$, $I_0$ and $C_0$ were adjusted; the microscopic state ($S$, $C$ or $P$) in each ensemble member was set as the state at the end of previous time step and was not adjusted. Detailed explanation of the IF-EAKF system is provided in Appendix 1.

In each iteration of the IF, the standard deviation of each parameter is shrunk by a factor $a \in (0, 1)$ (or equivalently, the variance is discounted by a factor of $a^2$). In practice, the discount factor $a$ can range between 0.9 and 0.99 (*Ionides et al., 2006*). If $a$ is too small, the algorithm may 'quench' too fast and fail to find the MLE; if it is too close to 1, the algorithm may not converge in a reasonable time interval. We stop the IF algorithm once the estimates of the ensemble mean stabilize. The number of iterations required for this convergence was determined by inspecting the evolution of posterior parameter distributions, as in *Figure 2A*. Note that once the ensemble mean stabilizes, increasing the iteration time will not affect the MLE, although it can lead to a further narrowing of the ensemble distribution.

Algorithm 1 only returns the MLEs for the parameters; however, it is also desirable to obtain CIs for those MLEs. For deterministic ODE models, Ionides *et al.* used 'sliced likelihood' to numerically estimate the Fisher information and standard errors (SEs) of MLEs (*Ionides et al., 2006*). Here, for a highly stochastic system, evaluating the Fisher information numerically is challenging. As a result, we took another approach by running multiple realizations of the IF algorithm. In different runs, the MLEs are slightly different due to stochasticity in the agent-based model and in the initialization of the inference algorithm. In this work, we ran 100 independent realizations to generate the average MLEs of inferred parameters and their corresponding 95% CIs. Results from synthetic tests indicate that this approach is effective in calculating MLEs and quantifying their uncertainties.

**Algorithm 1. IF–EAKF**

**Input:** An agent-based model $\mathcal{M}$ in a time-varying contact network $G(V, E, t)$, the number of observations $T$, incidence $\{o_t\}$, the observational error variance (OEV) $\{\sigma_{t,o}^2\}$, the initial system parameters $\bar{\mathbf{z}}^0 = (\beta, I_0, C_0)^T$, the initial covariance matrix $\Sigma$, a discount factor $a \in (0, 1)$, and the number of iterations $L$.
for $l = 1$ to $L$ do
 Generate an ensemble of parameter vectors with $n$ members using a multivariate Gaussian distribution: $\{\hat{\mathbf{z}}_0^l\}_n \sim \mathcal{N}(\bar{\mathbf{z}}^{l-1}, a^{2(l-1)}\Sigma)$.
 for $t = 1$ to $T$ do
 Run the agent-based model $\mathcal{M}$ with posterior $\{\hat{\mathbf{z}}_{t-1}^l\}_n$ obtained from last update for one week, and return the ensemble of incidence: $\{o_t^l\}_n = \mathcal{M}(G, \{\hat{\mathbf{z}}_{t-1}^l\}_n)$.
 Update the prior distribution of parameters $\beta$, $I_0$ and $C_0$ : $\{\mathbf{z}_t^l\}_n \equiv \{\hat{\mathbf{z}}_{t-1}^l\}_n$ to posterior $\{\hat{\mathbf{z}}_t^l\}_n$ using the EAKF, $\{o_t^l\}_n$, $\{\sigma_{t,o}^2\}$ and $\{o_t\}$. Individual states are evolved per the agent-based model and are not updated by the EAKF.
 end for
 Calculate the ensemble mean of the posterior over time as the input in next iteration: $\bar{\mathbf{z}}^l = \sum_t E(\{\hat{\mathbf{z}}_t^l\}_n)/T$, where $E$ computes the ensemble mean.
end for
**Output:** $\bar{\mathbf{z}}^L$ as the MLE of the parameter vector.

An alternative method to infer posterior parameters is to use Approximate Bayesian computation (ABC) (*Beaumont et al., 2002*). ABC-based methods employ numerical simulations to approximate the likelihood function, in which the simulated samples are compared with the observed data. In a typical ABC rejection algorithm, large numbers of parameters are sampled from the prior distribution. For each set of parameters, the distance between simulated samples (generated using the parameters) and observed data is calculated. Parameters resulting in a distance larger than a certain tolerance are rejected, and the retained parameters form the posterior distribution. ABC methods

can fully explore the likelihood landscape in parameter space. However, it requires large numbers of simulations, which may be prohibitive for the large-scale agent-based models considered here. In addition, a good choice of the tolerance in the rejection algorithm is needed. The IF algorithm, instead, is applicable to computationally expensive agent-based models, but may become trapped in the local optimum of the posterior distribution. In practice, this problem can be alleviated by exploring a larger prior parameter space and setting a slower quenching speed, that is, a smaller discount factor $a$.

## Inferred parameters and 95% CIs for three synthetic tests

We report the inferred parameters and their corresponding 95% CIs for the synthetic tests in *Table 3*. The actual parameters used to generate the synthetic outbreaks are also reported. Results are obtained from 100 independent realizations of the IF algorithm.

## Inference-based intervention

To guarantee a fair comparison between the inference-based intervention and other heuristic strategies, we estimated the colonization probability using only real-time information available before control measures are effected. For instance, to estimate the colonization probability at the fifth month in the third year, we first infer the model parameters for the first 2 years, where we have data from the whole year, and then use the partial observation in the remaining 5 months to infer the model parameters for the third year. The inferred parameters are then used to generate 1000 synthetic outbreaks from the beginning, and the current colonization probability for each individual is calculated from these simulations. In the inset of *Figure 5A*, we show that the colonization probability estimated in real time is highly correlated with that obtained using information from the entire outbreak record. In practice, every 4 weeks, the estimated colonization probability and the decolonization list were updated. The inference-based intervention only uses information available at the time control measures are effected. As a consequence, it is a practical method that can be implemented in real time.

## Acknowledgements

We thank Martin Bootsma, Ben Cooper, Prabhat Jha and Lulla Opatowski for valuable comments on earlier versions of the manuscript. JS is funded by the National Institute of General Medical Sciences GM110748 and the National Institute of Environmental Health Sciences ES009089. HM is funded by the National Institute of Biomedical Imaging and Bioengineering 1R01EB022720, the National Cancer Institute U54CA137788/U54CA132378 and the National Science Foundation IIS-1515022. The opinions stated are those of the authors and do not represent the official position of funders.

## Additional information

### Competing interests

Jeffrey L Shaman: Discloses partial ownership of SK Analytics. The other authors declare that no competing interests exist.

### Funding

| Funder | Grant reference number | Author |
| --- | --- | --- |
| National Institute of General Medical Sciences | GM110748 | Jeffrey L Shaman |
| National Institute of Environmental Health Sciences | ES009089 | Jeffrey L Shaman |
| National Institute of Biomedical Imaging and Bioengineering | 1R01EB022720 | Hernán Makse |
| National Cancer Institute | U54CA137788/U54CA132378 | Hernán Makse |

| National Science Foundation | IIS-1515022 | Hernán Makse |
|---|---|---|

The funders had no role in study design, data collection and interpretation, or the decision to submit the work for publication.

### Author contributions
Sen Pei, Conceptualization, Formal analysis, Investigation, Methodology, Writing—original draft; Flaviano Morone, Conceptualization, Investigation, Methodology, Writing—review and editing; Fredrik Liljeros, Conceptualization, Data curation, Investigation, Writing—review and editing; Hernán Makse, Jeffrey L Shaman, Conceptualization, Funding acquisition, Investigation, Methodology, Writing—review and editing

### Author ORCIDs
Sen Pei http://orcid.org/0000-0002-7072-2995
Jeffrey L Shaman http://orcid.org/0000-0002-7216-7809

### Ethics
Clinical trial registration
Human subjects: The dataset was approved for use in our study by the Regional Ethical Review Board in Stockholm (Record Number 2004/5:8).

### Decision letter and Author response
Decision letter https://doi.org/10.7554/eLife.40977.033
Author response https://doi.org/10.7554/eLife.40977.034

## Additional files

### Supplementary files
• Source code 1. The Matlab code for parameter inference in a synthetic MRSA outbreak simulated in an example time-varying contact network.
DOI: https://doi.org/10.7554/eLife.40977.030
• Transparent reporting form
DOI: https://doi.org/10.7554/eLife.40977.031

### Data availability
The dataset was approved for use in our study by the Regional Ethical Review Board in Stockholm (Record Number 2004/5:8). The data used in this article is a completely anonymized subset of the original data material, and keys to exact dates and identities of hospitals and individuals were not provided. While the data are de-identified, the owner of the data, Stockholm County Council, will not allow public sharing of the dataset. Interested researchers can verify the dataset at Stockholm University, and we, the authors, will do our best to help such interested researchers contact the owner of the data. Requests for data access should be addressed to FL (fredrik.liljeros@sociology.su.se). The Matlab code for parameter inference in a synthetic MRSA outbreak simulated in an example time-varying contact network is provided as Source code file 1. Numerical data for Figures 1-5 are also provided.

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

## Appendix 1

DOI: https://doi.org/10.7554/eLife.40977.032

## Analysis of the hospitalization traffic

We performed an analysis of the admission and discharge traffic in the study hospitals. In the hospitals, the inpatient population changed on a daily basis. To quantify the speed of patient renewal, we define a patient overlap ratio at day $t$ as $Q(t) = |Z(t-1) \cap Z(t)|/|Z(t-1)|$, where $Z(t)$ is the set of patients in the hospital at day $t$, and $|Z(t)|$ is the number of those patients. In **Figure 1—figure supplement 2A**, we present the evolution of $Q(t)$ during a period of 300 days. $Q(t)$ exhibits a somewhat periodic behavior with a period of 7 days (see the inset of **Figure 1—figure supplement 2A**). This is possibly due to reduced patient traffic during weekends.

We next examined the total number of patients in the study hospitals. As shown in **Figure 1—figure supplement 2B**, the in-hospital patient number fluctuates between 4000 and 7000. Patient numbers exhibit a periodic behavior at an annual time-scale, as well as at finer weekly time-scale. We also present the number of new patients (with respect to the patients present the previous day) in the hospitals each day in **Figure 1—figure supplement 2B**. The number of new patients is relatively small compared with the total patients.

The distribution of patient time in hospital follows a power-law shape with a heavy tail (see **Figure 1—figure supplement 2C**). Such heterogeneity leads to high spatiotemporal contact network complexity. In fact, the contact time between all pairs of patients follows a similar power-law distribution, as shown in the upper inset of **Figure 1—figure supplement 2C**. The readmission time, that is, individual patient time between discharge and next admission, is a key parameter in MRSA transmission models. The lower inset of **Figure 1—figure supplement 2C** indicates that this readmission time is also quite heterogeneous, spanning from several days to up to a few years.

Next we examined the topological features of the weekly aggregated contact network in **Figure 1—figure supplement 2D**, which shows the number of patients in the giant connected component (GCC) and the entire contact network. While most patients belong to the GCC, there also exist many fragmented small connected components (CCs). The total number of CCs in the contact network is also presented in **Figure 1—figure supplement 2D**. About 100 CCs coexist in the network each week, but the size of small CCs is usually below 200, as shown in the inset of **Figure 1—figure supplement 2D**. Connections between different CCs change over time due to the transfer and readmission of patients. 128,119 patients were transferred from one ward to another during their stay, and another 280,506 patients were readmitted within one year of their previous discharge. These patient movements connect healthcare facilities that would otherwise be isolated in the network and are responsible for long-range transmission across multiple hospitals.

Given the large heterogeneity in the network structure and contact time, a traditional compartmental model using ordinary differential equations (ODE) may not adequately capture actual transmission dynamics. Therefore, in this study we adopt an individual-level agent-based model.

## Equation-Free approach

Instead of using a parsimonious ordinary differential equation model, we employ an agent-based model to account for the spatiotemporal complexity of the underlying contact patterns. In particular, agent-based models can be used to simulate epidemic spread using an Equation-Free approach (**Kevrekidis et al., 2003**). The transmission process evolves following microscopic update rules defined at the individual-level, and macroscopic states are aggregated from the total simulated population.

The Equation-Free approach has been widely used for multi-scale modeling in applied mathematics and statistical physics. It consists of three basic elements: (1), lift, which transforms macroscopic observations through lifting to one or more consistent microscopic

realizations; (2), evolve, which uses the microscopic simulator to evolve these realizations for a given time; and (3), restrict, which aggregates the evolved microscopic realizations to obtain the macroscopic observation.

In the MRSA transmission model, some quantities, for example colonization importation, infection importation, and weekly incidence, are macroscopic values aggregated from the individual-level states. In model simulation, we first need to lift these macroscopic quantities to consistent microscopic realizations. To do this, we maintained multiple realizations (300 ensemble members) of individual-level states. For each new patient entering the hospitals, a random number $r$ was generated from a uniform distribution $v \sim U[0, 1]$. If $v \leq I_0$, $I_0 \leq v \leq I_0 + C_0$, or $v \geq I_0 + C_0$, and used to designate the new patient as infected, colonized or susceptible, respectively. This lifting procedure was performed for all realizations and produced an ensemble of possible microscopic states. These individual-level states were then evolved following the rules defined in the model over the time-varying contact network $G(V, E, t)$. The model estimate of the observed state, that is 4 week incidence, was obtained by aggregating the total number of new infections across the entire population in the study hospitals. This aggregated, macroscopic state is then used in conjunction with the EAKF algorithm to update the parameter vector $\mathbf{z} = (\beta, I_0, C_0)^T$ (see Materials and methods in main text). This multi-scale method enables system-level analysis directly from microscopic simulations, which bypasses the need to derive macroscopic evolution equations.

## The Ensemble Adjustment Kalman Filter

To represent the state-space distribution, the EAKF maintains an ensemble of system state vectors acting as samples from the distribution. In particular, the EAKF assumes that both the prior distribution and likelihood are Gaussian, and thus can be fully characterized by their first two moments, that is mean and covariance. The update scheme for ensemble members is computed using Bayes' rule (posterior $\propto$ prior $\times$ likelihood) via the convolution of the two Gaussian distributions. For observed state variables, the posterior of the $i^{\text{th}}$ ensemble member is updated through

$$o_{t,post}^i = \frac{\sigma_{t,obs}^2}{\sigma_{t,obs}^2 + \sigma_{t,prior}^2} \bar{o}_{t,prior} + \frac{\sigma_{t,prior}^2}{\sigma_{t,obs}^2 + \sigma_{t,prior}^2} o_t + \sqrt{\frac{\sigma_{t,obs}^2}{\sigma_{t,obs}^2 + \sigma_{t,prior}^2}} (o_{t,prior}^i - \bar{o}_{t,prior}). \tag{1}$$

Here $o_{t,post}^i$ and $o_{t,prior}^i$ are the posterior and prior of the observed variable for the $i^{\text{th}}$ ensemble member at time $t$; $\bar{o}_{t,prior}$ is the mean of the prior observed variable; $\sigma_{t,obs}^2$ and $\sigma_{t,prior}^2$ are the variances of the observation and the prior observed variable; and $o_t$ is the observation at time $t$. Unobserved variables and parameters are updated through their covariability with the observed variable, which can be computed directly from the ensemble. In particular, the $i^{\text{th}}$ ensemble member of unobserved variable or parameter $x^i$ is updated by

$$x_{t,post}^i = x_{t,prior}^i + \frac{\sigma(\{x_{t,prior}\}_n, \{o_{t,prior}\}_n)}{\sigma_{t,prior}^2} (o_{t,post}^i - o_{t,prior}^i). \tag{2}$$

Here $x_{t,post}^i$ and $x_{t,prior}^i$ are the posterior and prior of the unobserved variable or parameter for the $i^{\text{th}}$ ensemble member at time $t$; and $\sigma(\{x_{t,prior}\}_n, \{o_{t,prior}\}_n)$ is the covariance between the prior of the unobserved variable or parameter $\{x_{t,prior}\}_n$ and the prior of the observed variable $\{o_{t,prior}\}_n$ at time $t$. In the EAKF, variables and parameters are updated deterministically so that the higher moments of the prior distribution are preserved in the posterior.

## Synthetic tests

To generate synthetic outbreak observations, we used the agent-based model to simulate weekly incidence during a one-year period (52 weeks), and then imposed noise to produce the observations used in inference. The synthetic outbreak tested in *Figure 2* was generated with the parameters $\beta = 9 \times 10^{-3}$, $I_0 = 2 \times 10^{-3}$ and $C_0 = 7.5 \times 10^{-2}$. To mimic actual observational error variance, we assumed an observational error variance (OEV) of

$\sigma_{t,o}^2 = 10 + (0.1 \times o_{simulate}(t))^2$, which is a baseline uncertainty plus a term related to the simulated incidence $o_{simulate}(t)$. In reality, because we have only one data point at each observation time point, the variance of observed incidence is unknown. As such, we have to use a heuristic OEV in the inference algorithm. The above form of OEV has been successfully used in real-time influenza forecast (**Shaman and Karspeck, 2012**; **Pei and Shaman, 2017**; **Pei et al., 2018a**; **Kandula et al., 2018**), and also produces satisfactory performance in the following synthetic tests. The observation used in the inference was drawn from a Gaussian distribution $o(t) \sim \mathcal{N}(o_{simulate}(t), \sigma_{t,o}^2)$. The initial system state $\bar{\mathbf{z}}^0$ was drawn from the following ranges $\beta \in [0, 0.001]$, $I_0 \in [0, 0.003]$ and $C_0 \in [0, 0.1]$, using Latin Hypercube Sampling (LHS) (**Tang, 1993**). For simplicity, the initial covariance matrix was assumed to be diagonal $\Sigma = diag((\mathbf{z}_{max} - \mathbf{z}_{min})^2/16)$, where $\mathbf{z}_{max}$ and $\mathbf{z}_{min}$ are the vectors of the upper and lower bounds of parameters $\beta$, $I_0$ and $C_0$. In each iteration, the covariance matrix was contracted by a factor of $a^2$ (equivalent to a reduction of the standard deviation by a factor of $a$). We used a discount factor of $a = 0.9$ and terminated the algorithm at $L = 20$ iterations. For the EAKF, $n = 300$ ensemble members were used.

We validated the IF-EAKF inference framework for different synthetic scenarios. *Figure 2* presents the synthetic situation where nosocomial transmission accounts for the majority of incidence (see *Figure 2D–E*). To evaluate the goodness of fit for incidence number in *Figure 2B*, we performed the following statistical analysis. As the agent-based model is a highly stochastic system, the observed incidence in *Figure 2B* is only one possible outcome of the actual dynamics, whereas in our analysis, the stochasticity of incidence number needs to be considered. To this end, we compared several summary statistics quantifying the goodness of fit in *Figure 2B* with their distributions calculated from synthetic outbreaks (surrogate data) generated from the inferred dynamics. We first considered the log likelihood of observations. In particular, we generated 1000 synthetic outbreaks using the inferred parameters, and approximated the distribution of incidence number at each week. Then we calculated the log likelihood for the observed incidence in each synthetic outbreak, and estimated its distribution using these 1000 log likelihood values computed from the surrogate data. In *Figure 2—figure supplement 1A*, we compared the log likelihood computed from *Figure 2B* (vertical red line) with this distribution (blue bars) and calculated the 2-sided p-value. The p-value is well above zero, indicating that, in terms of log likelihood, our inferred dynamics span and thus agree well with the observed incidence. In other words, the observed incidence in *Figure 2B* is a typical outcome from our inferred dynamics. The same analysis was also applied to root-mean-square error (RMSE), coefficient of determination ($R^2$) and Pearson correlation coefficient (*Figure 2—figure supplement 1B–D*). The RMSE, $R^2$ and Pearson correlation coefficient were calculated using the incidence time series in each synthetic outbreak and the mean incidence time series averaged over 1000 simulations.

For the opposite situation in which nosocomial transmission is less than importation, we performed the same test. In this case, we set $\beta = 6 \times 10^{-3}$, $I_0 = 2 \times 10^{-3}$ and $C_0 = 7.5 \times 10^{-2}$. The distributions of posterior parameters after each iteration (blue boxes) shown in *Figure 2—figure supplement 2A* are gradually adjusted to their targets (red horizontal lines). We ran 100 independent realizations of the inference, and report the inferred values and 95% CIs for the parameters $\beta$, $I_0$ and $C_0$ in *Table 3*. Additionally, weekly incidence, colonized population, and nosocomial and imported infections can be generally reproduced with the inferred parameters (see *Figure 2—figure supplement 2B–E*). The goodness of fit in *Figure 2—figure supplement 2B* is analyzed in *Figure 2—figure supplement 3*.

We finally tested the effect of observation frequency. In the actual diagnostic data from the Swedish hospitals, weekly incidence is very low. To account for the large uncertainty in weekly observation, we instead use 4 week incidence. In *Figure 2* and *Figure 2—figure supplement 2*, we used weekly observations; in *Figure 2—figure supplement 4*, we assimilated 4 week incidence for a synthetic outbreak generated with the same parameter setting as in *Figure 2*. The findings indicate that this change of observational frequency does not affect the performance of the inference system. The inferred values and 95% CIs for the parameters $\beta$, $I_0$ and $C_0$ are reported in *Table 3*. The goodness of fit in *Figure 2—figure supplement 4B* is analyzed in *Figure 2—figure supplement 5*.

## Inference using the actual diagnostic data

Beginning with the first year, we ran the IF inference sequentially through each year. The initial ranges of parameters are set as $\beta \in [-, 0.01]$, $I_0 \in [0, 0.001]$ and $C_0 \in [0, 0.1]$. The initial covariance matrix is set as $\Sigma = diag((\mathbf{z}_{max} - \mathbf{z}_{min})^2/16)$, where $\mathbf{z}_{max}$ and $\mathbf{z}_{min}$ are the vectors of the upper and lower bound of parameters $\beta$, $I_0$ and $C_0$. In EAKF, the OEV is defined as $\sigma_{t,o}^2 = 1 + (0.1 \times o_t)^2$, where $o_t$ is the observed incidence at time $t$. In each year, we ran $L = 20$ iterations using a discount parameter of $a = 0.9$. Between two consecutive years, we used the ensemble of inferred parameters and microscopic states of individuals in the previous year to initialize the inference system in the next year in order to maintain the continuity of the transmission dynamics. The evolution of the posterior parameter distributions for 20 iterations through the 6 year record is shown in *Figure 3—figure supplement 1*. In general, the ensemble means of parameters become stable after 10 iterations with the EAKF through the record, which means our choice of $L = 20$ is sufficient. Note that increasing $L$ will lead to narrower distributions of posterior parameters in *Figure 3—figure supplement 1*, but will not affect the MLEs, that is the ensemble mean. We repeated the inference 100 times and obtained the inferred parameters and corresponding 95% CIs reported in *Table 2*. We further performed an evaluation of the goodness of fit in *Figure 3—figure supplement 2*. All statistics have p-values well above zero. This implies the observed incidence in *Figure 3B* is a plausible outcome from the inferred dynamics.

Following inference, we used the mean estimated parameters to simulate outbreaks. Key information, for example incidence (*Figure 3B*), the infections transmitted in hospital and imported from outside the study hospitals (*Figure 3E*), the colonized population (*Figure 4A*), and the individual-level colonization rate (*Figure 4B–C*) were obtained from these simulations.

## Statistical test of power-law distribution

We used a maximum likelihood estimator to fit and validate the power-law data in *Figure 4B*. For computational convenience, we fit the count of infections of each individual among $10^4$ simulations (that is, we multiply the frequency in *Figure 4B* by $10^4$). Denote the count data by $X = (x_1, x_2, \cdots, x_n)$, and suppose the data satisfy a power-law distribution $P(x_i) \propto x_i^{-\gamma}$. Usually, empirical data follow a power-law behavior above a lower bound: $P(x_i) \propto x_i^{-\gamma}$ for $x_i \geq x_{min}$ (*Clauset et al., 2009*). For a given $x_{min}$, the maximum likelihood estimator of the power-law exponent $\gamma$ for discrete data is $\hat{\gamma} = 1 + n[\sum_{i=1}^n \ln(x_i/x_{min} - 1/2)]^{-1}$ (*Clauset et al., 2009*). The standard error on $\hat{\gamma}$ is estimated by $\sigma = (\hat{\gamma} - 1)/\sqrt{(n)} + O(1/n)$. To find the best lower bound, $x_{min}$ ranging from 0 to 100 was tested. The best $x_{min}$ was selected by choosing the value that minimizes the Kolmogorov-Smirnov (KS) statistic between the data and the fitted model. KS statistic is the maximum distance between two cumulative distribution functions (CDFs): $D = \max_{x \geq x_{min}} |S(x) - P(x)|$, where $S(x)$ is the CDF of the data larger than $x_{min}$, and $P(x)$ is the CDF of the power-law model obtained from MLE. In *Figure 4—figure supplement 1A*, the change of KS statistic for different $x_{min}$ values is reported. We choose the best $x_{min}$ value as 24. The fitted power-law exponent is $\hat{\gamma} = 2.13$ with a standard deviation of 0.12.

To test the statistical significance of the power-law fitting, we generated $10^4$ synthetic datasets using the fitted model, computed their KS statistic with respect to the model, and compared the obtained distribution with the KS statistic of the data in *Figure 4—figure supplement 1B*. If the p-value of the observed KS statistic is close to 0, we reject the hypothesis that the data are generated from the fitted model; otherwise, the power-law fitting is statistically significant. This approach has been widely used in testing the statistical significance of power-law fitting (*Clauset et al., 2009*; *Muchnik et al., 2013*). The 2-sided p-value for our data is 0.5374.

