## [Decision Letter]

Thank you for submitting your article "Inference and control of the nosocomial transmission of Methicillin-resistant Staphylococcus aureus" for consideration by *eLife*. Your article has been reviewed by three peer reviewers, including Ben Cooper as the Reviewing Editor and Reviewer #1, and the evaluation has been overseen by a Reviewing Editor and Prabhat Jha as the Senior Editor. The following individuals involved in review of your submission have agreed to reveal their identity: Martin Bootsma (Reviewer #2); Lulla Opatowski (Reviewer #3).

The reviewers have discussed the reviews with one another and the Reviewing Editor has drafted this decision to help you prepare a revised submission.

Summary:

This work makes both an important methodological contribution to the communicable disease modelling literature, and also an important practical contribution to the study of nosocomial infections. First, the authors develop a novel inference framework allowing inferences from individual based transmission models (overcoming computational problems with existing particle filtering approaches). Second, the methods are applied to make inferences about parameters for a multi-institutional MRSA outbreak across a network of hospitals in Sweden and simulations are used to show how such model-based inferences could translate into improved control measures for MRSA.

Essential revisions:

1) Clarification of exact nature of the MRSA data is needed. Does this only come from clinical infections or was there also screening on asymptomatic carriage?

2) Figure 1B is difficult to read and should be revised. Since only integers (0-5) are plotted, the continuous scale may not be the best choice and makes it difficult to pick out what numbers are being plotted.

3) Enough information needs to be presented to allow an interested reader to repeat this analysis on their own data. In particular, the following aspects of the methods were not clear to any of the reviewers:

i) In Algorithm 1, why is the discount factor raised to the power 2(l-1), rather than just (l-1). Is the 2 arbitrary?

ii) How should the discount factor *a* be chosen, and how do different values of *a* influence the algorithm? A value of 0.9 is used, but not explanation is given for this choice.

iii) Algorithm 1 returns the MLE of the parameter vector but makes no attempt to measure the associated uncertainty. Indeed, the variance of the parameter vector ensemble approaches zero as L increases. This is also seen in Appendix 1, Figure 4A. However, distributions for parameters are reported in Figure 3A, and the methods say that "parameter distributions become stable after 5 iterations…which means our choice of L=10 is sufficient". So how are the distributions in Figure 3A derived? Are these just taken from the ensemble of parameter vectors stopping after 10 iterations (implying that if L had been taken as 20 or more, the distributions shown in Figure 3A would have had much lower variance?) Surely it is important to produce not only point estimates, but also appropriate measures of uncertainty associated with these; it is not clear if this is being done and justification of the choice of L=10 in the data analysis (subsection “Inference using the actual diagnostic data”) compared with L=20 in the synthetic tests (subsection “Synthetic tests”) is lacking.

iv) In Algorithm 1 the actual EAKF step is not explained. Given that space is not a limitation (and technical details can be put int he appendix), it would be helpful to provide details of this EAKF step with sufficient detail to enable replication by the interested reader.

4) Clarification of model assumptions is needed. In particular, it is unclear whether the increased decolonization rate *μ* was only present during treatment, and that afterwards the spontaneous decolonization rate *α* kicks in, or whether treated patients keep their increased rate *μ* also after the treatment is stopped. Linked to this, what is the duration of treatment?

5) An explicit unambiguous description of the transmission process within a ward is lacking. Is the process density-dependent or frequency-dependent?

6) Units are missing in Table 1.

7) A table with all parameters estimates is missing.

8) Figure 5A-B legend is confusing as the y-axis doesn't seem to depict a reduction but cumulative incidence. Clarification is needed.

9) Can more formal assessment of model fits with synthetic and real data be provided in addition to the plots in Figure 2 and Figure 3?

10) Figure 5 and associated text: it is unclear whether the model based control measures uses only data/information that would be available at the time point control measures are put in place. If the inference approach instead uses information from the whole course of the epidemic, then this would be an unfair comparison. This needs clarification.

11) Subsection “Statistical test of power-law distribution”: "the significance level of 0.05" Following the recent ASA statement on the use (and misuse of p-values) there is a consensus that we should be striving to move away from such "bright lines". So fine to report the p-value, but we should try to avoid saying that. 049 is significant while. 051 isn't.

---

## [Author Response]

Summary:This work makes both an important methodological contribution to the communicable disease modelling literature, and also an important practical contribution to the study of nosocomial infections. First, the authors develop a novel inference framework allowing inferences from individual based transmission models (overcoming computational problems with existing particle filtering approaches). Second, the methods are applied to make inferences about parameters for a multi-institutional MRSA outbreak across a network of hospitals in Sweden and simulations are used to show how such model-based inferences could translate into improved control measures for MRSA.

We appreciate the reviewers’ positive evaluations of our work.

Essential revisions:1) Clarification of exact nature of the MRSA data is needed. Does this only come from clinical infections or was there also screening on asymptomatic carriage?

Diagnosis was performed on patients with symptomatic infections as well as some patients with contact with positive cases. So, the screening covers both clinical infections and a portion of asymptomatic carriage. This is now clarified in subsection “Data description” of the revised manuscript.

2) Figure 1B is difficult to read and should be revised. Since only integers (0-5) are plotted, the continuous scale may not be the best choice and makes it difficult to pick out what numbers are being plotted.

Figure 1B was reproduced using a discrete color palette.

3) Enough information needs to be presented to allow an interested reader to repeat this analysis on their own data. In particular, the following aspects of the methods were not clear to any of the reviewers:i) In Algorithm 1, why is the discount factor raised to the power 2(l-1), rather than just (l-1). Is the 2 arbitrary?

After each iteration, the standard deviation of each parameter is discounted by a factor *a*. This is equivalent to a discount of *a*^2^ on the variance, which is represented by multiplication with the diagonal matrix Σ. We have clarified this point in subsection “Iterated filtering for agent-based models” and subsection “Synthetic tests” of the revised manuscript.

ii) How should the discount factor a be chosen, and how do different values of a influence the algorithm? A value of 0.9 is used, but not explanation is given for this choice.

In implementation, the discount factor, *a*, is typically selected to be between 0.9 and 0.99 (Ionides et al., 2006). If it is too small, the algorithm may “quench” too fast and fail to find the MLE; if it is too close to 1, the algorithm may not converge in a reasonable time interval. We now explain the choice of discount factor in subsection “Iterated filtering for agent-based models”.

iii) Algorithm 1 returns the MLE of the parameter vector but makes no attempt to measure the associated uncertainty. Indeed, the variance of the parameter vector ensemble approaches zero as L increases. This is also seen in Appendix 1, Figure 4A. However, distributions for parameters are reported in Figure 3A, and the methods say that "parameter distributions become stable after 5 iterations…which means our choice of L=10 is sufficient". So how are the distributions in Figure 3A derived? Are these just taken from the ensemble of parameter vectors stopping after 10 iterations (implying that if L had been taken as 20 or more, the distributions shown in Figure 3A would have had much lower variance?) Surely it is important to produce not only point estimates, but also appropriate measures of uncertainty associated with these; it is not clear if this is being done and justification of the choice of L=10 in the data analysis (subsection “Inference using the actual diagnostic data”) compared with L=20 in the synthetic tests (subsection “Synthetic tests”) is lacking.

This is an insightful question. Indeed, Algorithm 1 only returns the MLEs of parameters. For deterministic ODE models, Ionides et al. used “sliced likelihood” to numerically estimate the Fisher information and standard errors (SEs) of the MLEs (Ionides et al., 2006). Here, for a highly stochastic system, evaluating the Fisher information numerically is challenging. As a result, we employed another approach by running multiple realizations of the IF algorithm. In these different runs, the MLEs are slightly different due to the stochasticity in the agent-based model and the initialization of the inference algorithm. We ran 100 independent realizations to generate the average MLEs of inferred parameters and their corresponding 95% CIs. We explain this approach in subsection “Iterated filtering for agent-based models”.

The distributions shown in Figure 3A of the initial submission were the posterior distributions of 300 ensemble members after L=10 iterations in one realization of the IF algorithm. The reviewers are correct that if we continue the algorithm, these distributions will converge to a point, as is observed in Figure 2A and supplemental figures. To properly report the inference results in the revision, we now display the distributions of MLEs (ensemble means) obtained from the 100 runs in Figure 3A. In other figures, we have also clarified which distributions have been plotted. Please see changes in subsection “Iterated filtering for agent-based models”, subsection “Inference of MRSA transmission in Swedish hospitals” and relevant figure legends.

We stopped the algorithm after the ensemble mean estimate stabilized. In principle, the number of iterations required for convergence can be determined by inspecting the evolution of posterior parameter distributions, as in Figure 2A. Once the posterior mean stabilizes, increasing the iteration time will not affect the MLE, although it will surely lead to a gradually narrowed ensemble distribution. This point is now explained in subsection “Iterated filtering for agent-based models”. To keep the algorithm setting consistent with the synthetic test, we have extended the iteration time to L=20 for the inference using real-world diagnostic data in the revision.

iv) In Algorithm 1 the actual EAKF step is not explained. Given that space is not a limitation (and technical details can be put int he appendix), it would be helpful to provide details of this EAKF step with sufficient detail to enable replication by the interested reader.

A detailed explanation of the EAKF update was added to the appendix. Please see subsection “The Ensemble Adjustment Kalman Filter”.

4) Clarification of model assumptions is needed. In particular, it is unclear whether the increased decolonization rate μ was only present during treatment, and that afterwards the spontaneous decolonization rate α kicks in, or whether treated patients keep their increased rate μ also after the treatment is stopped. Linked to this, what is the duration of treatment?

We apologize for this ambiguity. In our model, we assume that treatment continues until infected patients are clear of MRSA. Given the exponential decay of infection probability, the characteristic treatment period is 1/μ days. We have clarified this assumption in subsection “The agent-based model” of the revised manuscript.

5) An explicit unambiguous description of the transmission process within a ward is lacking. Is the process density-dependent or frequency-dependent?

We now explain the transmission process explicitly in subsection “The agent-based model” of the revised manuscript. Specifically, a susceptible individual staying in a ward with a colonized person can become MRSA colonized with transmission probability β per day. In our model, we assume that patients within a ward have the same rate of contact with each other, presumably mediated by the shared healthcare workers in a ward. The transmission process is density-dependent, as the force of infection in a ward increases with the number of colonized patients within the ward.

6) Units are missing in Table 1.

Units were added in Table 1.

7) A table with all parameters estimates is missing.

All estimated parameters are now reported in Table 2.

8) Figure 5A-B legend is confusing as the y-axis doesn't seem to depict a reduction but cumulative incidence. Clarification is needed.

The legend of Figure 5A-B was changed to indicate that the y-axis is cumulative incidence.

9) Can more formal assessment of model fits with synthetic and real data be provided in addition to the plots in Figure 2 and Figure 3?We thank reviewers for this constructive suggestion. For the revision, we performed an analysis to evaluate the model fit. To do so, we inspect whether the observed incidence is a typical outcome from the inferred stochastic dynamics. Specifically, we compared several summary statistics quantifying the goodness-of-fit with their distributions calculated from synthetic outbreaks (surrogate data) generated from the inferred dynamics. The tested statistics include log likelihood, RMSE, R-squared and Pearson correlation coefficient. For all statistics, p-values are bounded away from zero, indicating that the observed incidence agrees well with the inferred dynamics. Details can be found in subsection “Synthetic tests”, Figure 2—figure supplement 1, Figure 2—figure supplement 3, Figure 2—figure supplement 5 and Figure 3—figure supplement 2.10) Figure 5 and associated text: it is unclear whether the model based control measures uses only data/information that would be available at the time point control measures are put in place. If the inference approach instead uses information from the whole course of the epidemic, then this would be an unfair comparison. This needs clarification.

To guarantee a fair comparison with other control measures, we reran the inference-based intervention using only real-time information available before the control measures are effected. The colonization probability estimated in real-time is highly correlated with the results obtained using information from the whole course of the epidemic. The implementation details are now presented in the revised manuscript and can be found in subsection “Designing cost-effective interventions” and subsection “Inference-based intervention”. The results remain similar.

11) Subsection “Statistical test of power-law distribution”: "the significance level of 0.05" Following the recent ASA statement on the use (and misuse of p-values) there is a consensus that we should be striving to move away from such "bright lines". So fine to report the p-value, but we should try to avoid saying that.049 is significant while.051 isn't.

Statements about statistical significance were removed.